# A Novel Spatio-Temporal FCN-LSTM Network for Recognizing Various Crop Types Using Multi-Temporal Radar Images

**Nima Teimouri \*, Mads Dyrmann** and **Rasmus Nyholm Jørgensen**

Department of Engineering-Signal Processing, Faculty of Science and Technology, Aarhus University, 8200 Aarhus N, Denmark; madsdyrmann@eng.au.dk (M.D.); rnj@eng.au.dk (R.N.J.)

\* Correspondence: n.teimouri@eng.au.dk; Tel.: +45-91789213

**Abstract:** In recent years, analyzing Synthetic Aperture Radar (SAR) data has turned into one of the challenging and interesting topics in remote sensing. Radar sensors are capable of imaging Earth's surface independently of the weather conditions, local time of day, penetrating of waves through clouds, and containing spatial information on agricultural crop types. Based on these characteristics, the main goal sought in this research is to reveal the SAR imaging data capability in recognizing various agricultural crops in the main growth season in a more clarified and detailed way by using a deep-learning-based method. In the present research, the multi-temporal C-band Sentinel 1 images were used to classify 14 major classes of agricultural crops plus background in Denmark. By considering the capability of a deep learning method in analyzing satellite images, a novel, optimal, and lightweight network structure was developed and implemented based on a combination of a fully convolutional network (FCN) and a convolutional long short-term memory (ConvLSTM) network. The average pixel-based accuracy and Intersection over Union obtained from the proposed network were 86% and 0.64, respectively. Winter rapeseed, winter barley, winter wheat, spring barley, and sugar beet had the highest pixel-based accuracies of 95%, 94%, 93%, 90%, and 90%; respectively. The pixel-based accuracies for eight crop types and the background class were more than 84%. The network prediction showed that in field borders the classification confidence was lower than the center regions of the fields. However, the proposed structure has been able to identify different crops in multi-temporal Sentinel 1 data of a large area of around 254 thousand hectares with high performance.

**Keywords:** long-short term memory; fully convolutional network; sequential network; multi-temporal data; crop classification; Sentinel 1

## 1. Introduction

The increase in population pressure across the world and thus the demand for the increase in agricultural crops has led to demands for improving agricultural resource management globally. To do so, first, obtaining reliable information—not only on the quantity, but also on the quality and place of those resources—is a vital necessity. Satellite and geographic information system (GIS) data serve as a highly important factor in improving the current systems of the collection and development of agricultural maps and data sources. Freely available satellite data constitute one of the most applied sources for mapping agricultural land and assessing important indices that describe the conditions of fields [1]. Analyzing such data makes it possible to provide accurate insights in man–environment interactions; specifically, multi-spectral and radar satellite image analysis could be an effective and accurate decision-making tool for the user. Satellites with optic or radar sensors provide images

globally with different time and place division powers. Those satellites have been used for various agricultural-related analyses including evaluation of the health and development status of crops, planning irrigation, creating agricultural crops maps, and analyzing soil [2–7].

By using machine learning-based semantic image segmentation, the class of each pixel is determined [8]. In general, machine learning is divided into supervised and unsupervised training categories. In the supervised training of semantic segmentation algorithms, the true class and location of pixels in the training data should be known beforehand. In an agricultural context, this knowledge could be gained from a combination of field surveys, the interpretation of aerial images, or simply reports of what has been planted or sown in fields. In contrary, in the unsupervised classification of satellite data, the type and place of various land coverage which is to be determined is not known in advance; thus, there are no label or reference data [9]. One of the main reasons for using unsupervised methods in remote sensing is a lack of annotated reference land data.

Many studies have already been carried out by using optic sensors such as Landsat 8, MODIS, and Sentinel 2 satellites in the agricultural domain for assessing important key factors such as crop height, biomass activity, soil moisture, leaf area index (LAI), and mapping land cover [1,10–13]. In contrast to optical systems, Synthetic Aperture Radar (SAR) satellites have their own source of radar radiation; these waves are divided into different categories including the X band (2.5–3.75 cm), C band (3.75–7.5 cm), and L band (15–30 cm). One of the reasons that have made SAR data popular among researches in different areas is their invulnerability to different weather conditions, penetrability of clouds, and capability of obtaining data both day and night [14]. A number of research works have been carried out which use radar images from resources such as ALOS-PALSAR (Japan), TerraSAR-X (Germany), Sentinel 1 (European Space Agency), and Radarsat 1 and 2 (Canada); however, more comprehensive and detailed research is needed in the agricultural domain as well. Erten et al. (2016) [15] studied SAR data to assess the vegetation height parameter. For monitoring agricultural crops and studying soil conditions, SAR data were employed with interesting results [16]. Radar images have been used to predict biomass activity and leaf area indexes, and the results of these studies have indicated a correlation between radar backscatter and these indices [17–19]. Also, the C-band Sentinel 1 SAR data have been analyzed temporally to recognize which agricultural crops grow in fields [20–23]. Likewise, Kussul et al. (2017) [24] identified maize, wheat, sunflower, sugar beet, and soybean fields by analyzing a combination of multi-spectral and SAR data using two different multi-layer perceptron (MLP) and convolutional neural networks (CNNs). In another study, single data analysis of radar scans from Sentinel 1 was used to classify four different agricultural crops [25]. Satellite scans have also been used to classify plants using support vector machines (SVMs) [26,27], random forest (RF) [28,29], and multiscale convolutional neural network [30]. In the past two years, recurrent neural networks (RNN) for analyzing multi-temporal data have received attention from researchers in remote sensing [31,32]. A number of studies have already been carried out, which analyzed optical and radar images using RNNs to gain better results as well [21,33]; however, those results indicate that it is still a challenge to develop a robust and generalizable method to recognize various agricultural crops using only C-band Sentinel 1 data.

Even though the literature includes multiple studies dealing with crop-classification from C-band SAR images, those studies are limited in terms of covered area, considered crop-types, and utilization of temporal information. Therefore, the aim of the present research is to develop a system based on a hybrid of a fully convolutional network (FCN) and a convolutional long short-term memory (LSTM) network (ConvLSTM), which, together, analyze C-band Sentinel 1 images and identify 14 different crop types plus background.

## 2. Materials

The data used in the present study were obtained from the Sentinel 1 satellite with two polarizations: VV and VH. The pixel size of the orthorectified image was 10 m. These images require calibration and preprocessing before analysis including terrain correction, radiometric correction for creating



radar backscatter according to digital pixel values, speckle, and thermal noise removal. All data were acquired from Fieldbabel (https://fieldbabel.eng.au.dk), a web service developed by Aarhus University to do all the preprocessing steps mentioned above [7]. In Figure 1, the red colored zones were selected for training the network, covering 504 thousand hectares, and the blue colored zones, covering 254 hectares, were used for evaluating the network (Table 1). Since the area used for evaluating the network varies geographically and in terms of soil and crop types, the results are believed to be generalizable to the rest of Denmark. It is observed that the satellite data were collected from specific geographic regions because of the distribution of various agricultural crop types in these areas in Denmark. The images, which were analyzed from 1 May 2017 to 31 August 2017, cover the main growth season for different crops in Denmark.

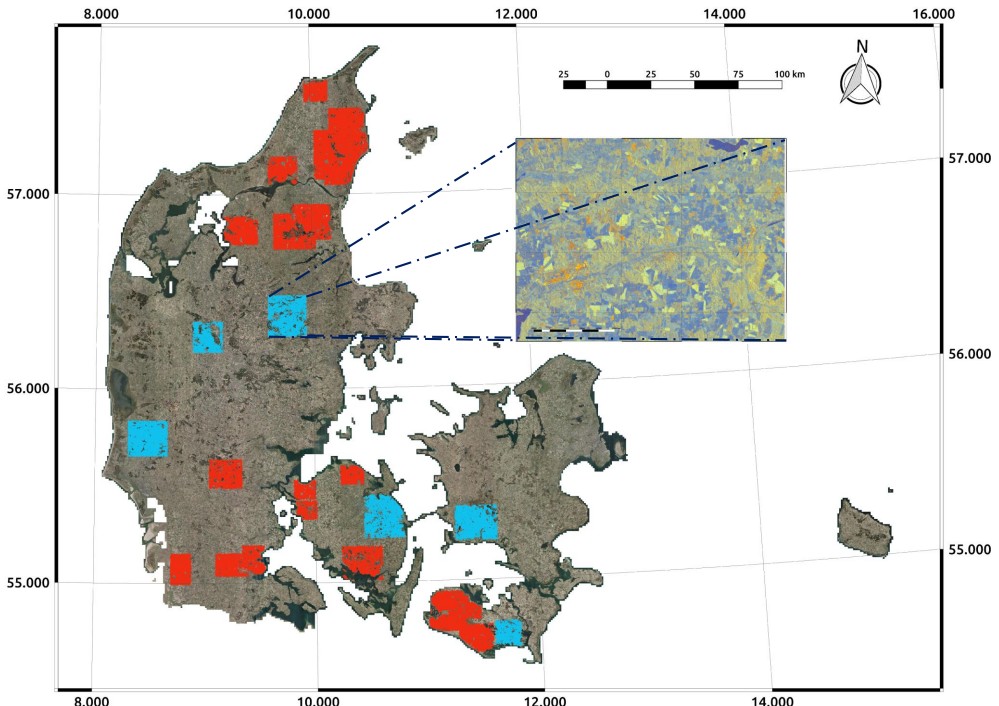

**Figure 1.** Training and test regions in Denmark (the red and blue color zones represent training and test regions).

**Table 1.** The distribution of agricultural crop types in 2017.

| ID | Class | Calibration Area (ha) | Test Area (ha) | Test Area (%) |
|----|-------|----------------------|----------------|---------------|
| 1 | Winter barley | 16,904.94 | 7526.98 | 2.95 |
| 2 | Winter wheat | 82,524.95 | 35,219.85 | 13.83 |
| 3 | Winter rye and hybrid rye | 20,425.98 | 4727.83 | 1.86 |
| 4 | Winter triticale | 2198.21 | 1010.08 | 0.40 |
| 5 | Winter rapeseed | 23,035.48 | 11,688.5 | 4.59 |
| 6 | Spring barley | 81,121.04 | 39,712.37 | 15.59 |
| 7 | Spring oats | 10,704.71 | 2480.07 | 0.97 |
| 8 | Maize | 25,238.24 | 12,486.71 | 4.90 |
| 9 | Sugar beet | 16,003.96 | 3470.58 | 1.36 |
| 10 | Potato | 84,710.84 | 4424.42 | 1.74 |
| 11 | Green grain spring barley | 3303.15 | 661.11 | 0.26 |
| 12 | Peas | 2411.76 | 1751.12 | 0.69 |
| 13 | Grass. seed | 5392.92 | 1837.69 | 0.72 |
| 14 | Permanent grass | 24,767.93 | 8707.79 | 3.42 |
| 15 | Background | 23,438.09 | 84917.7 | 33.34 |
| 16 | Other crops | 82,546.88 | 34,101.06 | 13.39 |
| Total | | 504,729.08 | 254,723.86 | 100.00 |

All steps of image processing and deep learning methods were carried out in Python using open source computer vision (OPENCV) and TensorFlow library.

## 3. Methods

### 3.1. The Structure of CNN

Convolution neural networks (CNNs) are composed of feed-forward neural networks in which several layers are trained in a robust and optimized way. In 2012, Krizhevsky et al. [34] demonstrated how a CNN could increase the classification accuracy significantly in the ImageNet challenge [34]. Since then, all leading image classification methods in the ImageNet challenge are based on convolutional networks [35–37]. In general, a CNN consists of three main layers; namely, the convolution layer, the pooling layer, and the fully connected layers. Different layers perform various functions. There are two stages for training in each convolutional network; the feed-forward stage and the backpropagation state.

In the first step, each neuron receives a number of inputs, then proceeds with calculating the products of calculated weights in the entries and, at the end, the output value is obtained by using a non-linear activation function [38]. The main function of the convolution layer in the network is the extraction of spatial features and the pooling layers decrease the dimensions of feature maps to obtain a faster training procedure. A CNN provides a differentiable score function, one side of which contains the raw pixels of the input image; the scores of each class are on the other side of it. So, the network output is used to calculate the loss for the purpose of training the network. In other words, the network output is compared with the actual response by using a loss function, and the differentiation shows the loss. Finally, the backpropagation stage is commenced based on the calculated loss value. In this stage, the gradient of each parameter is calculated and all parameters are updated as per the amount of loss. The feed-forward step starts after updating the parameters. After a suitable repetition of those stages, the network training stage ends and the final weights of the network are saved for evaluation on the test set.

### 3.2. Our Approach

The aim of this study is to analyze SAR data temporarily and spatially. For this purpose, a combination of a recurrent neural network (RNN) and an FCN is used. In combination, these two networks are denoted ConvLSTM. The next section describes a ConvLSTM cell briefly.

LSTM is an improvement over RNN, which is utilized approach to process the sequential data with relatively long term dependencies and possess a vanishing gradient problem. A typical ConvLSTM unit contains inputs, gates, outputs, and memory cell (Figure 2). The Equations (1)–(6) formally describe the gates and outputs of ConvLSTM cell.

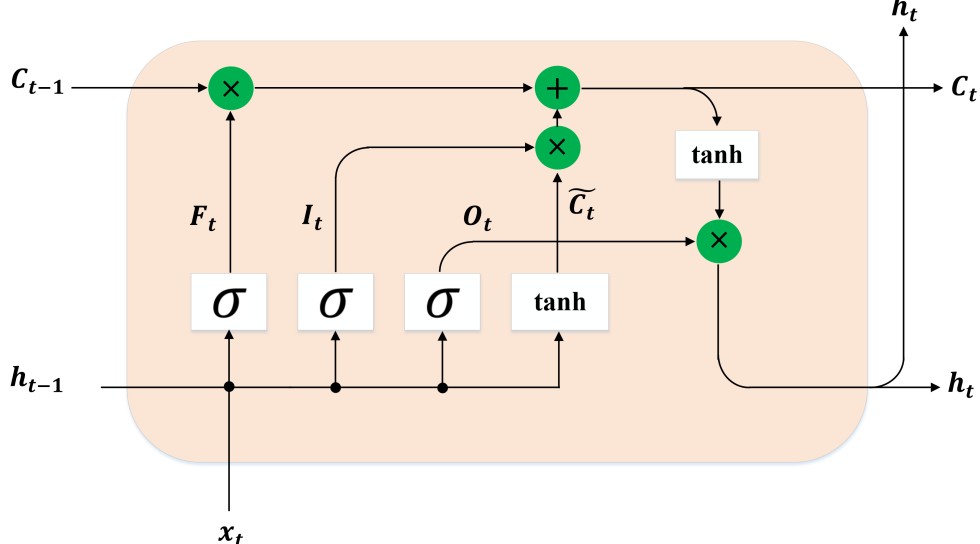

**Figure 2.** Schematic illustration of a convolutional long short-term memory (ConvLSTM) cell with all variables.

$$I_t = \sigma(W_{ix} * x_t + W_{ih} * h_{t-1} + b_i) \tag{1}$$

$$F_t = \sigma\left(W_{fx} * x_t + W_{fh} * h_{t-1} + b_f\right) \tag{2}$$

$$O_t = \sigma(W_{ox} * x_t + W_{oh} * h_{t-1} + b_o) \tag{3}$$

$$\widetilde{C_t} = \tanh(W_{cx} * x_t + W_{ch} * h_{t-1} + b_c) \tag{4}$$

$$C_t = \widetilde{C_t} \odot I_t + C_{t-1} \odot F_t \tag{5}$$

$$h_t = \tanh(C_t) \odot O_t \tag{6}$$

In Figure 2 and Equations (1)–(6), $x_t \in R^d$ is the current input which $t \in \{1, \ldots, T\}$ is the sequence of observations, T is the maximum length of observations, and d is the input depth, $h_{t-1} \in R^{h \times w \times r}$ is the output of the previous ConvLSTM block which $h$ and $w$ are height and width of images, and $r$ is the number of recurrent cells, $C_{t-1} \in R^{h \times w \times r}$ is the output of the previous ConvLSTM block cell memory, $h_t \in R^{h \times w \times r}$ is the current block output, and, finally, $C_t \in R^{h \times w \times r}$ is the memory output of the current cell. $W \in R^{(r+d) \times r}$ are recurrent weights adjusted during the training process and $b$ are the bias terms. $I_t$, $F_t$, $O_t$, and $\widetilde{C_t}$ are input, forget, output, and candidate gates, respectively. Also, $\sigma$ and tanh are the logistic sigmoid and hyperbolic tangent, mapping real numbers to (0,1) and (−1,1), respectively. In the above mathematical equations, $*$ and $\odot$ denote 2D spatial convolution operation and pixel-wise product. Depending on the inputs to each block, each unit makes its decision based on the current input, the output of the previous unit, and the unit's previous memory; it then creates a new output and modifies its memory. Considering the advantages and capabilities of ConvLSTM units, a novel, optimal, and light sequential network combining FCN and ConvLSTM is suggested in this research for sequential data representation. The schematic of the proposed network is illustrated in Figure 3. The network is made up of two main components, Main blocks 1 and 2, which both contain convolutional layers, pooling layers, and ConvLSTM units (Figure 4). Main block 1 takes as input sequential images with a resolution of 96 × 96 pixels. During training and testing 28 were used, but the architecture is able to handle an arbitrary number of images. In the first block the network has two 3 × 3 convolutional layers, followed by a 2 × 2 max-pooling layer. In the second and third blocks, there are two and three 3 × 3 convolutional layers that both are followed by 2 × 2 max-pooling layers. We used a skip connection in the network to preserve features that are extracted in the initial layers and are

required for reconstruction in the up-sampling layer. Following Main block 1 is Main block 2, which contains two $3 \times 3$ ConvLSTM layers. In the recurrent cells, the hyperbolic tangent activation function was used to create the output based on the sequential input data. The score map was obtained from the output of the last ConvLSTM layer and a Softmax layer with same number of neurons as classes was stacked on the last LSTM unit to predict the final multi-class. The Softmax function is a generalization of a logistic function that squashes a k-dimensional vector z of arbitrary real values to a k-dimensional $\sigma = z$ of real values in the range [0, 1] that add up to 1. It should be noted that developing a lightweight network, which are only about 64 MB in total, is one of the main advantages of this study.

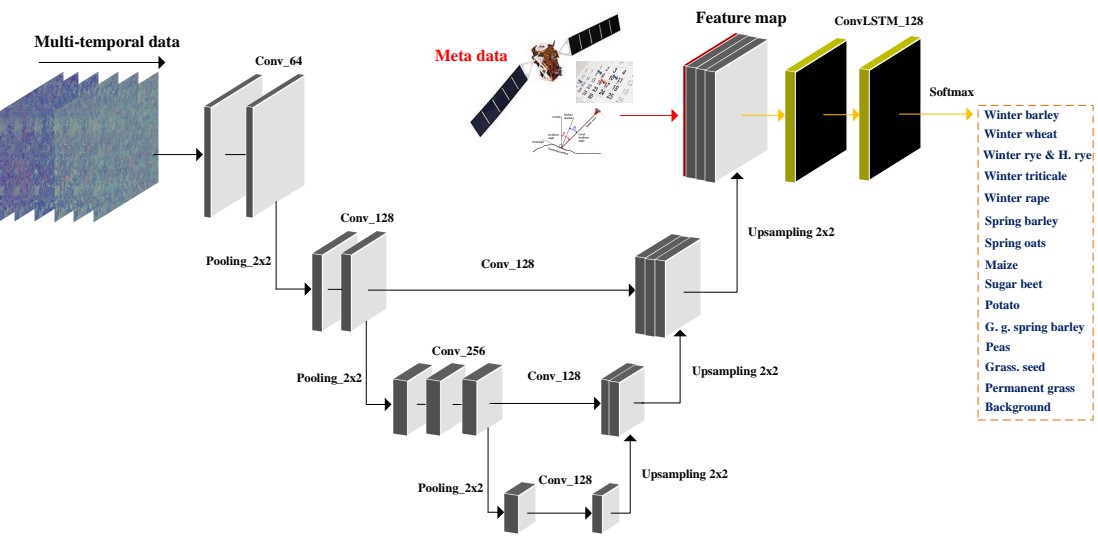

**Figure 3.** Overall architecture of proposed sequential network for recognizing different crop types.

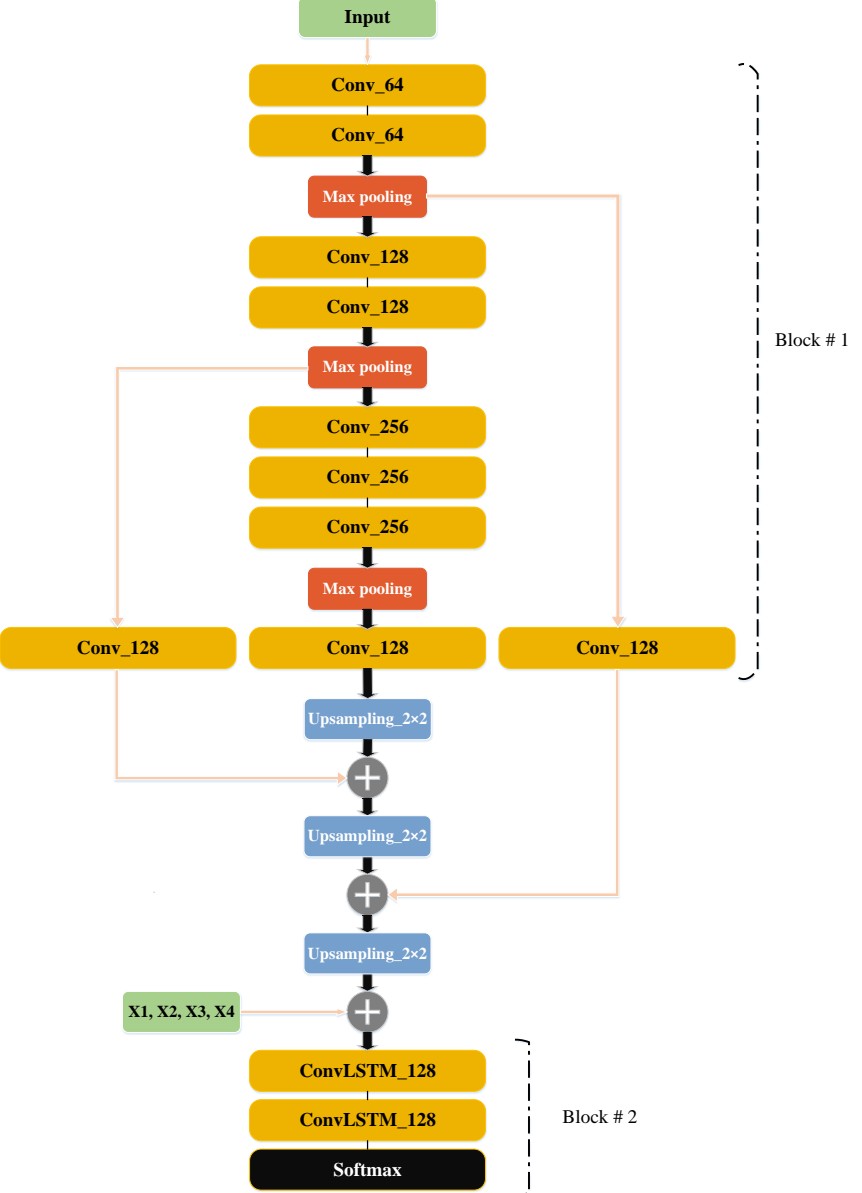

**Figure 4.** Illustration of our approach based on the combined fully convolutional network (FCN) and ConvLSTM network.

The proposed network can extract the spatiotemporal features successfully using the two main blocks (Figure 4). The first block includes a series of convolution and pooling layers to process input tensors (VV, VH, and VV/VH Sentinel 1 images) and to extract appropriate spatial information such as color and texture properties from different crops in radar images. Also, the second main block consists of ConvLSTM layers for extracting temporal features from multi-temporal images. The input data for the second block are composed of the output of the first block, day and hour of recording images, and incidence near and far angles which all of them normalized between 0 and 1. There were 28 observation dates between May 2017 and August 2017 and all 28 images were fed as one mini-batch into the network in order to train the model based on the different development stages belonging to the classes defined in this research. It should be noted that the shape files were provided by Land-parcel identification system (LPIS), which has an error level about 2.9%. This error level is quite low, enabling us to create a reference image for all input image sequences captured from Sentinel 1 satellite [39].

*3.3. Training*

The network was trained from scratch using the Adam optimizer with a learning rate of 0.0001 and a weight decay of 0.0001 per iteration. Each iteration consists of only one sample, which is a time-series that contains 28 images. The loss is based on binary cross-entropy, which is the difference between the predictions and a one-hot representation of the label-images. However, since some classes are harder to recognize than others, focal loss is added to make easily-recognizable pixels contribute less to the loss than hard-to-recognize pixels [40]. Finally all parameters are updated based on the calculated error. Each training sample consists of a 96 × 96 pixel patch, which is randomly cropped from the high-resolution training areas from Figure 1. To increase the generalizability and robustness of the network, the training images and labels were augmented with random scaling (zooming) and rotation (Figure 5).

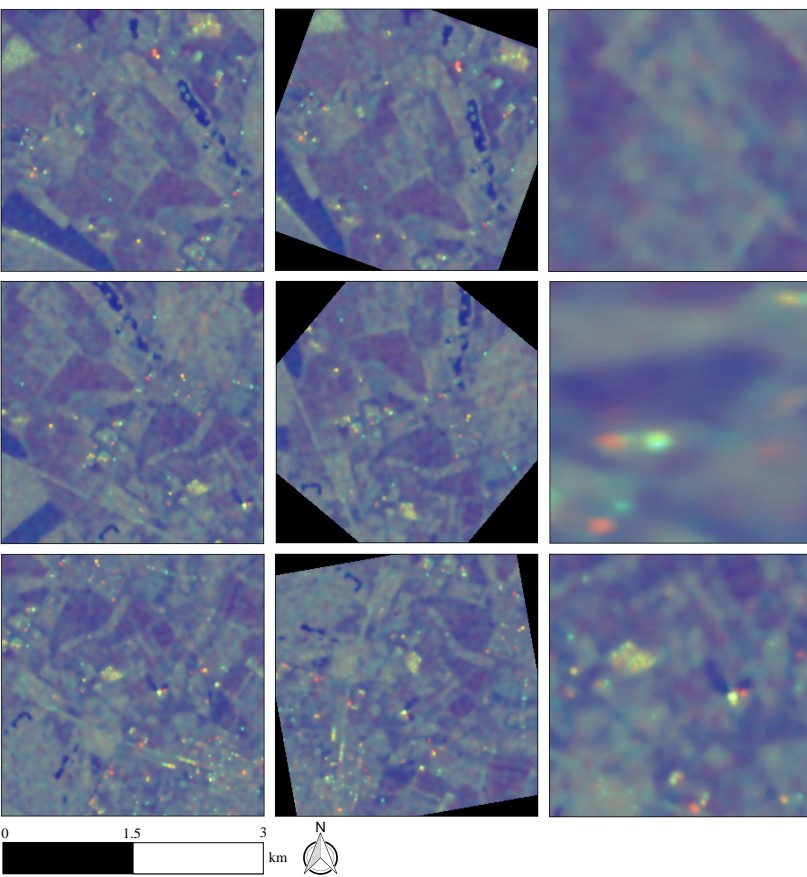

**Figure 5.** Visualization of augmented images using rotating and scaling.

*3.4. Accuracy Assessment*

Classification accuracies for 14 crop types were calculated using label-images obtained from an annually updated shapefile of all crops grown in Denmark. The pixel-based accuracy and Intersection over Union (IoU) [41] were used to evaluate the network's performance.

## 4. Results and Discussions

*4.1. Comparison Between Two Structures of LSTM Units*

For the evaluation of the proposed approach, we trained the whole network from scratch twice by using 128 or 256 ConvLSTM cells, respectively. First, 3 × 3 kernel sizes were employed for the evaluation because previous results with larger kernel sizes demonstrated approximately similar accuracies.

The results showed that 256 recurrent cells achieved slightly better accuracies compared with the method using only 128 cells. However, the training and evaluations times increased significantly, while the validation loss only dropped from 0.074 to 0.071 when going from 128 to 256 cells. We, therefore, went with 128 recurrent cells, as we believe fewer parameters helps the network to be more generalizable. The proposed network was trained on an NVidia GTX 1080.

## 4.2. Quantitative Classification Evaluation

The total test area covers around 254 thousand hectares including the background class. Of this area, background, spring barley, and winter wheat cover 84,917, 39,712, and 35,219 hectares, respectively (Table 1). Thereby, those three classes take up around 62.76% of the total test area. In addition, the areas covered by all 15 classes except green grain spring barley were larger than 1000 hectares. The area covered in this study is, therefore, more than 96 times bigger than that used in [21], which demonstrates the generalizability of our approach. Figure 6 shows a normalized confusion matrix of the predictions on the test set. The average pixel-based accuracy is 86%, showing the success of the proposed method in recognizing 15 different classes using multi-temporal C-band SAR Sentinel 1 images.

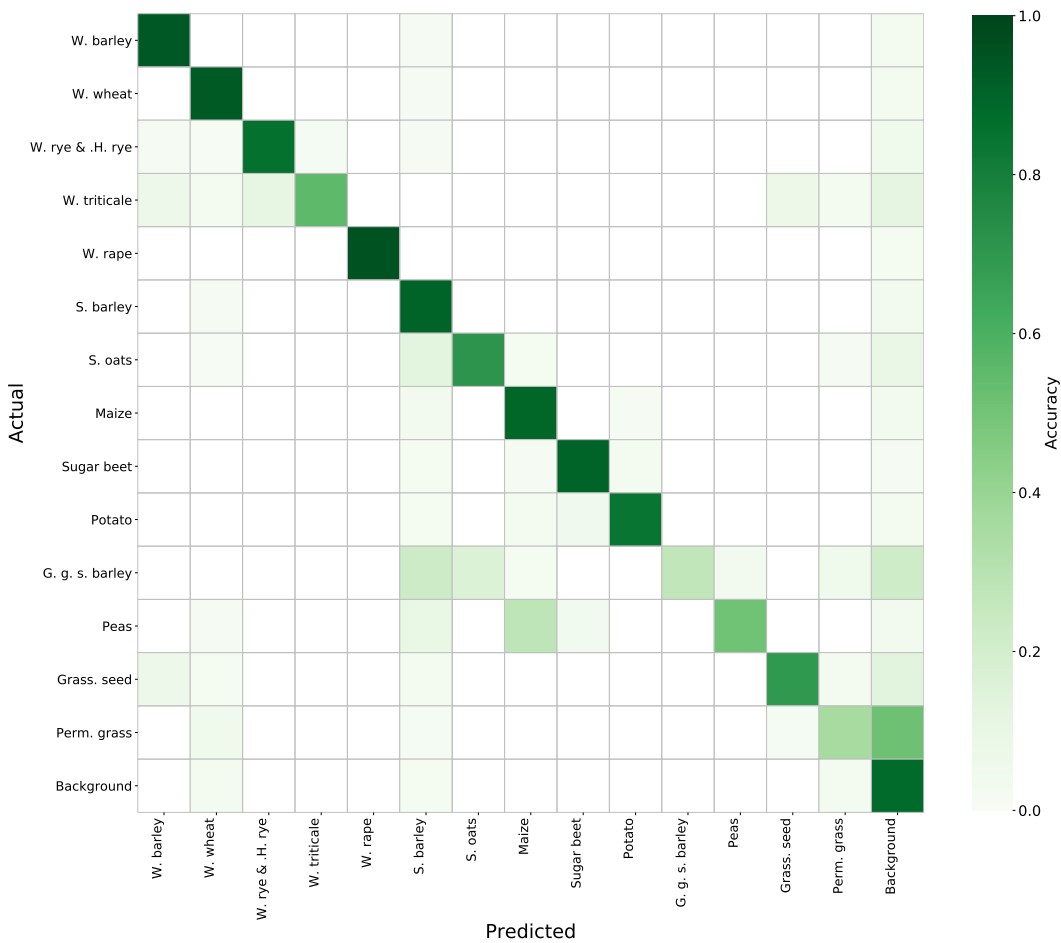

**Figure 6.** Normalized confusion matrix obtained from the proposed network with 15 different classes.

The highest accuracies were achieved for winter rapeseed, winter barley, winter wheat, spring barley, and sugar beet, achieving 95%, 94%, 93%, 90%, and 90% accuracies, respectively. The network, however, had trouble classifying green grain spring barley and permanent grass correctly. Green grain spring barley was mostly confused with spring barley, which makes sense since the only difference between the two is the time of harvest. The only difference is, therefore, the last images from the season. Permanent grass is mostly confused with the background class, which is believed to be caused by the

fact that land areas not registered as farmland have permanent grass. Thereby, the background class also contains permanent grass, which confuses the classifier.

One of the important and interesting classes is the background class, which contains all regions that are not registered as farmland such as forest, buildings, roads, lakes, and sea. The pixel-based accuracy for the background class is 88%. One of the reasons for the high accuracy of this class, despite its high intra-class variability, could be the difference in temporal development of the background compared to the other classes. Hence, the pixel intensity of the different crops undergoes essential changes in the four-month main growth period; no such changes were found for background class pixels in the same period of time. The positive impact of temporal information can be seen in Figure 7, where the performance of our approach in July and August was significantly better than that in May.

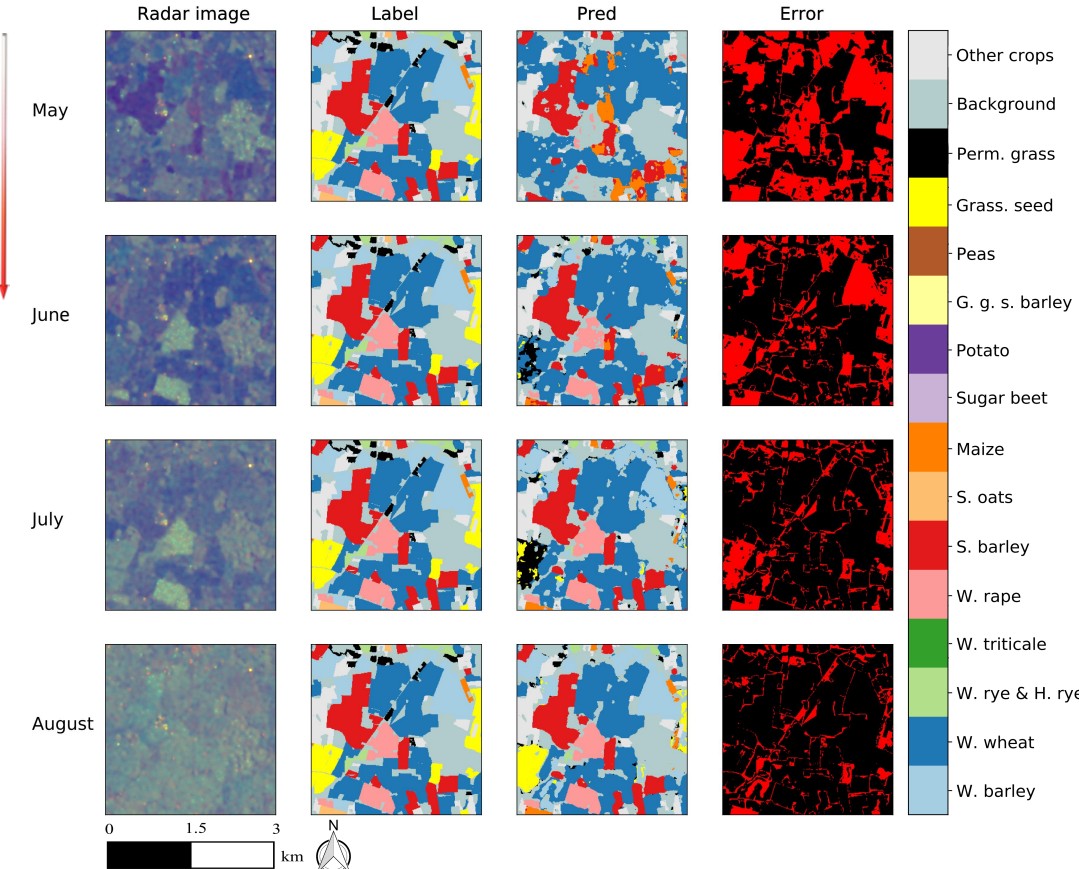

**Figure 7.** Qualitative results of the proposed sequential network (red pixels in the Error column represent the error).

The reason for the better performance at the end of the growth season is probably due to more temporal information. From the error column in Figure 7, it appears that limited information in the Sentinel 1 data was provided for the model before June, which is in line with the phenology of the various crop types that are approximately similar in the early growing season. Therefore, the information available from Sentinel 1 in the early months is not sufficient for the model to separate all the different classes from each other with high accuracy. However, winter wheat, spring barley, and winter rapeseed can be distinguished in May. Furthermore, by adding multi-temporal data from June, new crops such as winter barley, maize, and sugar beet were predicted with more than 86% pixel-based accuracy as compared with the remaining classes (Table 2 and Figure 8).

In Table 2, it is shown that the IoU for winter barley and potato, respectively, were 0.42 and 0.09 in May. These figures improved significantly in June to 0.73 and 0.68, respectively. The phonological

profiles for winter barley and wheat were close in May, but in the next month phonological changes for these two crops started to differ. Thus, the model could separate winter barley from winter wheat well when including the data in June (Figures 8 and 9). Another interesting class is spring oats because the accuracy of this crop was only 1% by the end of May, but reached 71% in July (Table 2). Overall, the accuracy and IoU indices for all classes did not change significantly from the end of July till the end of August (Figures 8 and 9). Therefore, by using only the multi-temporal data from May, June, and July, the proposed model could recognize 15 different classes with the accuracies presented in this study and we could omit multi-temporal images in August from the training procedure (Table 2).

**Table 2.** The performance when analyzing multi-temporal images. In the table Acc and IoU are pixel-based accuracy and Intersection over Union, respectively.

| Classes | May | | June | | July | | August | |
|---|---|---|---|---|---|---|---|---|
| | Acc (%) | IoU | Acc (%) | IoU | Acc (%) | IoU | Acc (%) | IoU |
| Winter barley | 60 | 0.42 | 91 | 0.73 | 93 | 0.81 | 94 | 0.79 |
| Winter wheat | 87 | 0.68 | 92 | 0.82 | 93 | 0.83 | 93 | 0.85 |
| Winter rye and hybrid rye | 55 | 0.43 | 75 | 0.63 | 85 | 0.74 | 86 | 0.74 |
| Winter triticale | 3 | 0.03 | 24 | 0.21 | 51 | 0.39 | 55 | 0.40 |
| Winter rapeseed | 85 | 0.77 | 96 | 0.90 | 95 | 0.90 | 95 | 0.90 |
| Spring barley | 83 | 0.66 | 85 | 0.75 | 91 | 0.82 | 90 | 0.82 |
| Spring oats | 1 | 0.01 | 32 | 0.14 | 71 | 0.53 | 71 | 0.51 |
| Maize | 67 | 0.35 | 86 | 0.66 | 89 | 0.74 | 89 | 0.75 |
| Sugar beet | 77 | 0.64 | 91 | 0.80 | 91 | 0.79 | 90 | 0.79 |
| Potato | 10 | 0.09 | 79 | 0.68 | 84 | 0.73 | 84 | 0.74 |
| G. g. spring barley | 1 | 0.01 | 3 | 0.03 | 27 | 0.24 | 27 | 0.24 |
| Peas | 16 | 0.14 | 47 | 0.41 | 49 | 0.44 | 51 | 0.45 |
| Grass. seed | 14 | 0.13 | 64 | 0.51 | 74 | 0.61 | 70 | 0.57 |
| Permanent grass | 27 | 0.20 | 30 | 0.23 | 34 | 0.27 | 36 | 0.29 |
| Background | 86 | 0.71 | 87 | 0.75 | 88 | 0.77 | 88 | 0.77 |

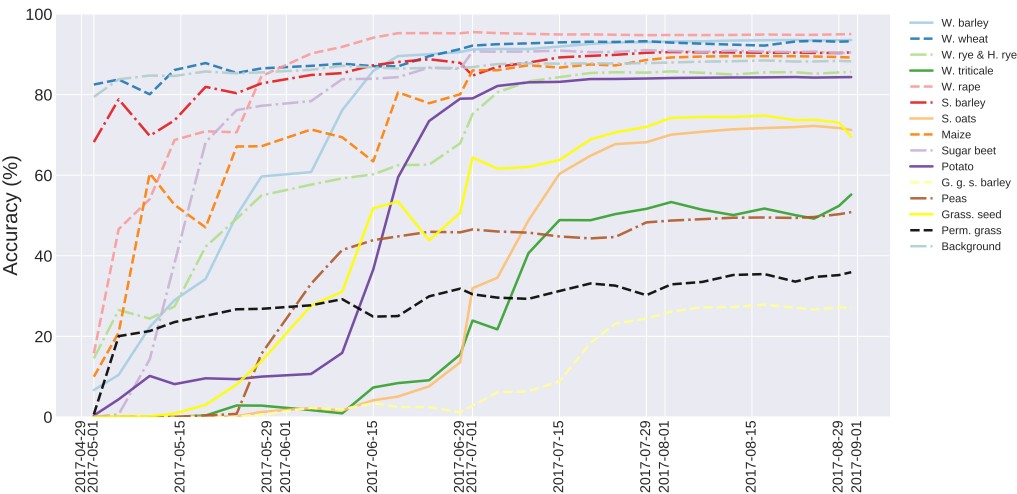

**Figure 8.** The pixel-based accuracy at different dates.

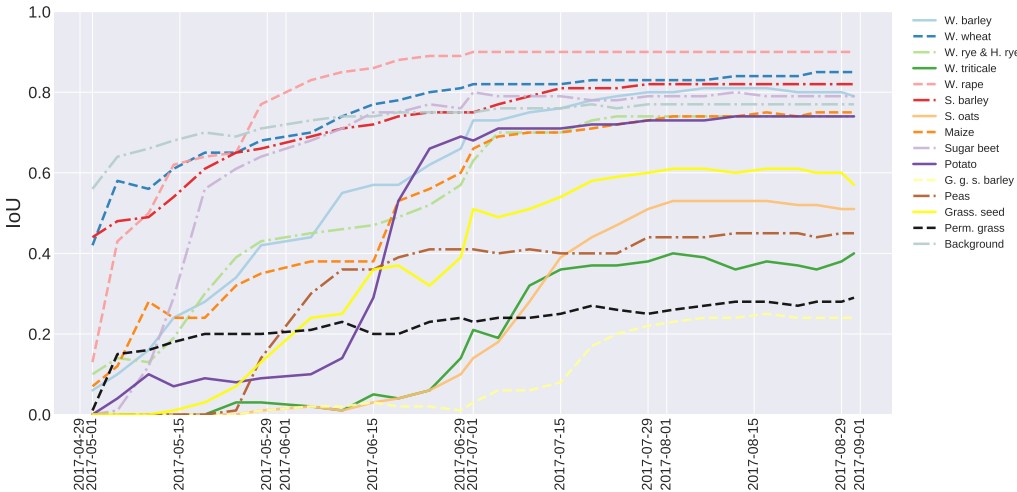

**Figure 9.** The IoU index at different dates.

Figure 10 shows the IoU for all classes. This index is extensively used in the semantic segmentation and indicates the degree of overlap between the reference image and the output of the network; thus, the IoU alongside the pixel-based accuracy reveals the performance of the proposed network clearly. The mean value of the IoU index was found to be 0.64; however, as it can be seen in Figure 10, the IoU varied for different classes based on the amount of complexity or the degree of dissimilarity within each class. Winter rapeseed with 0.90 had the highest IoU; green grain spring barley had the lowest IoU of 0.24. Only a few fields of green grain spring barley were predicted correctly and this crop was thus the most complicated case in the current research (Figure 10). This could be because green grain spring barley has a growth pattern similar to spring barley; furthermore, the harvest of this crop was conducted only two weeks before the harvest of the spring barley crop.

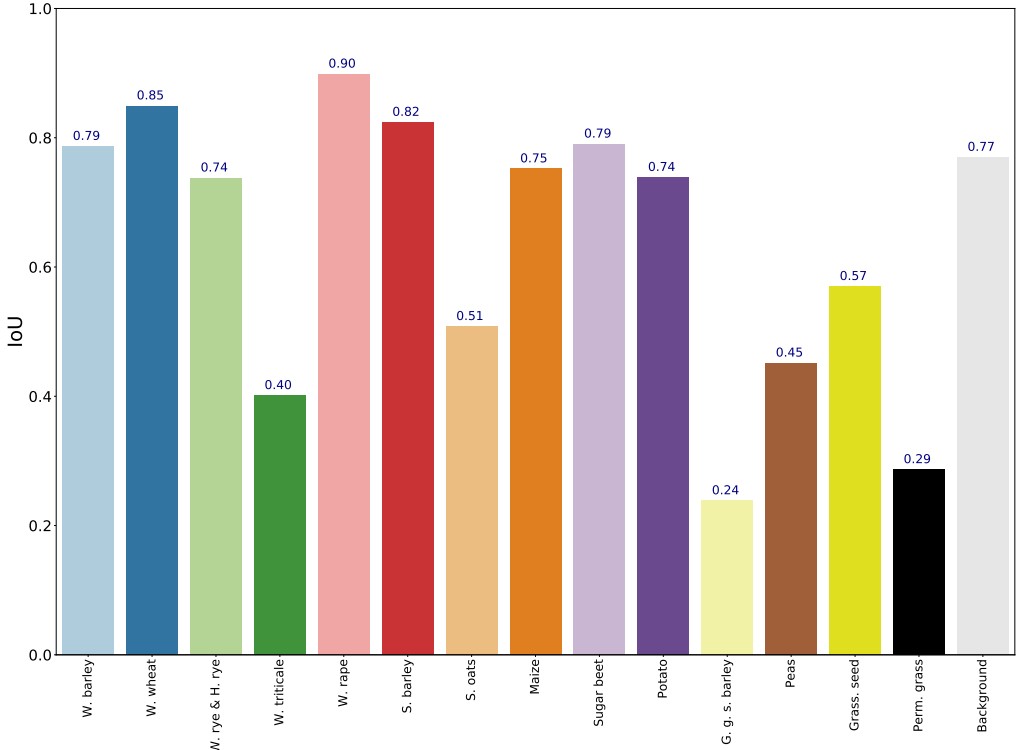

**Figure 10.** The IoU of all analyzed crops by the end of August.

Winter triticale is a hybrid of winter wheat and winter rye, and it can be seen in the confusion matrix as well. The largest number of misclassified pixels belong to these two classes. Therefore, there are reasonable causes for the rather poor ability of the network to separate winter triticale from other crops.

From Figures 11 and 12, it appears that the confidence is lowest in the boundary area of the different fields while there are fewer misclassified pixels inside the fields compare with the borders. Similar results have been reported by the literature [33]. The main errors of our approach were in the boundary regions of fields (Figures 11 and 12) and only a few fields were classified incorrectly, which could be due to field dissimilarity. The network's errors in the field boundaries are logical as the reference images might not be fully aligned with the fields. Moreover, the spatial resolution of 10 m/px also introduced noise from shelter-belts or neighboring fields in the boundary areas. Similar results were reported by [33]. During the preprocessing steps of the Sentinel 1 data, different types of errors appeared, so we cannot expect to accurately extract the field borders from the reference images.

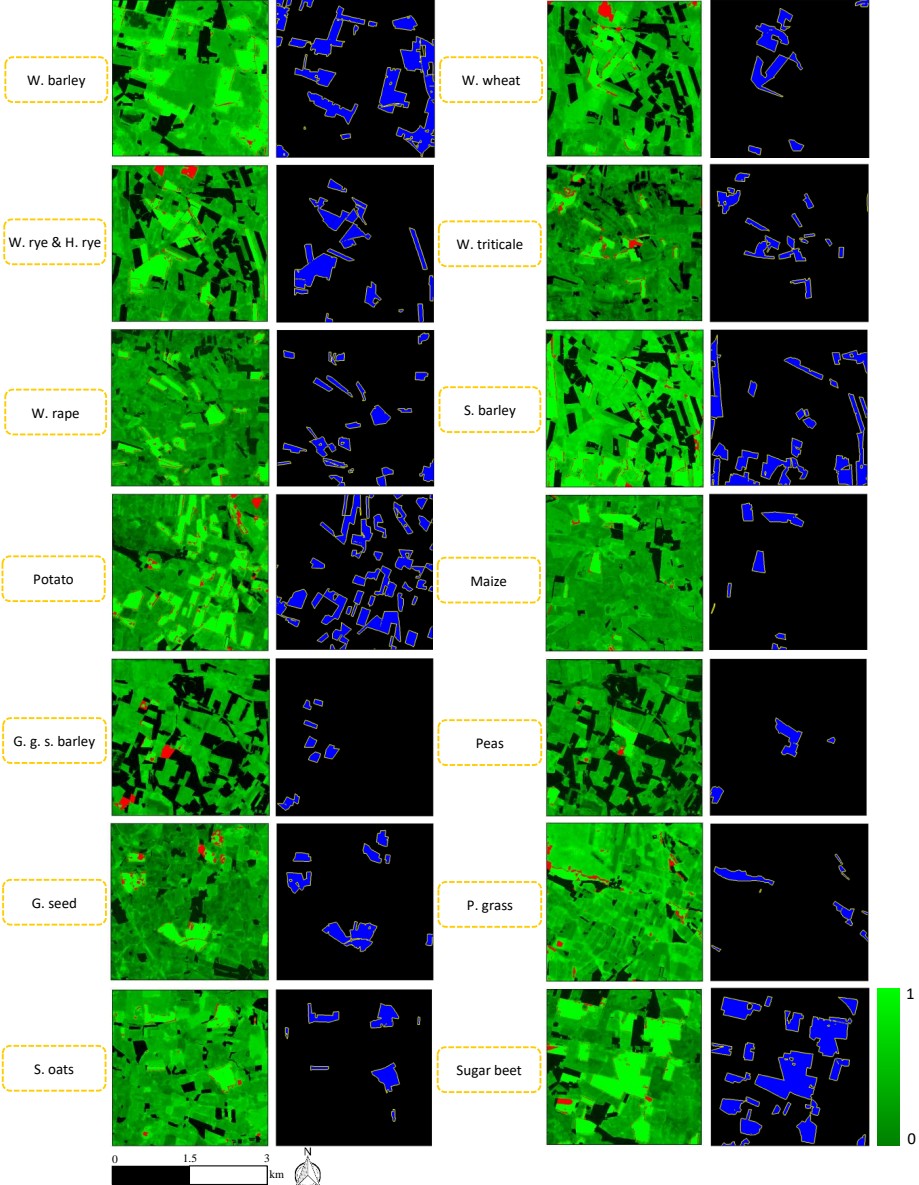

**Figure 11.** Classification confidence of the proposed method. Red pixels and blue regions correspond to the error and label images, respectively. The green color bar shows the confidence of our approach for each crop type.

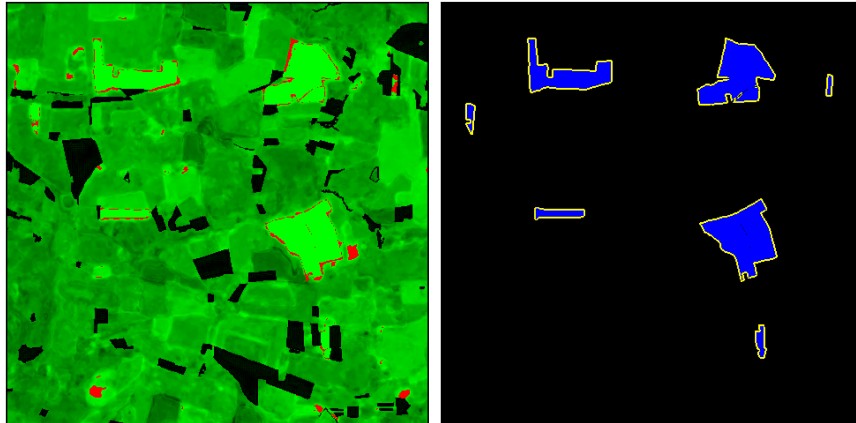

**Figure 12.** The classification confidence and the errors of the proposed method for spring oats. The color bar and scale are the same as in Figure 11.

One of the main advantages of the proposed network is the identification of fields that are likely annotated incorrectly in the reference images. As an example, in Figure 13, three different fields in the reference image were annotated as winter rapeseed, spring oats, and permanent grass, while our approach classified them as maize, spring barley, and background, respectively, with high confidence and sharp field boundaries. Thus, we may be able to improve the performance of the network if all the reference data are annotated correctly (Figure 13).

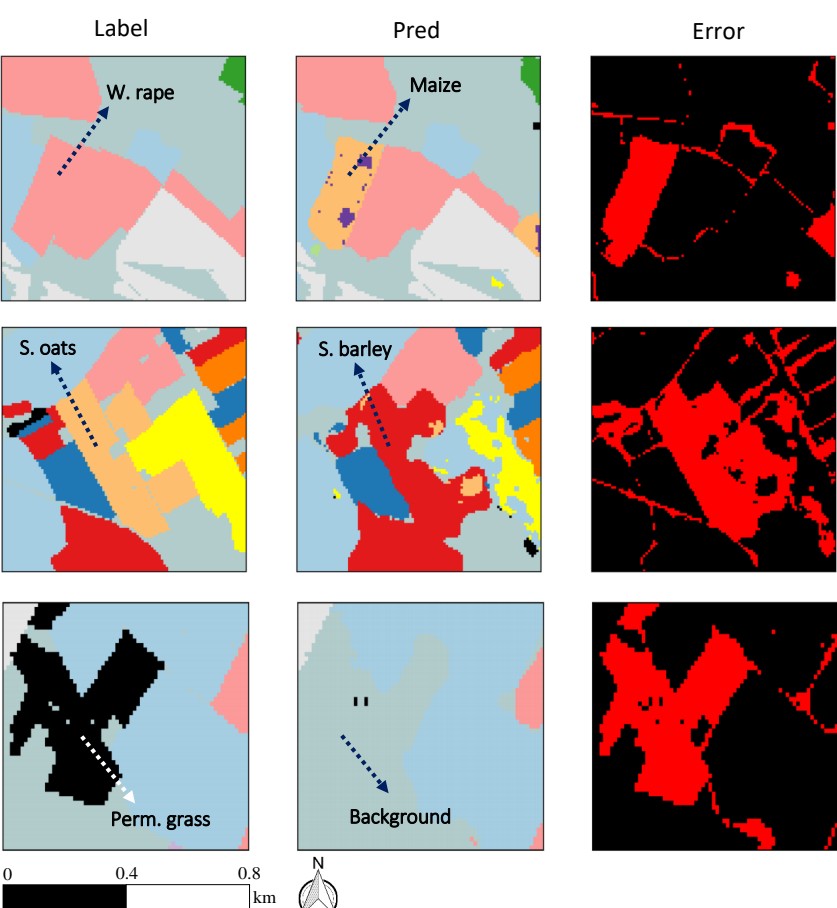

**Figure 13.** Some areas which were classified with high confidence by the proposed network, whilst these fields likely had wrong label in the reference images. The color legends are similar to Figure 7.

Overall, by evaluating the results obtained from our approach, we can conclude that this method is able to identify 15 different classes of agricultural crops (including background) from each other with 86% accuracy. Moreover, in the future work, the goal is to utilize both Sentinel 1 and 2 images to study the capability of combining SAR and multi-spectral data to recognize 14 crop types and background using our approach.

### 4.3. Comparison with Alternative State-Of-The-Art Methods for Recognizing Crop Types

Table 3 shows some perspective of the current research in relation to other studies that analyzed multi-temporal Sentinel 1 images with or without optical sensors for the purposes of mapping land cover. However, our method, which utilizes only Sentinel 1 data, achieved higher accuracies for the various crop types when analyzing 254 thousand hectares. With regard to [21], they almost achieved the same accuracy, but with fewer classes. Therefore, we implemented their network and trained it using our dataset with all 15 classes. The mean pixel-based accuracy was 71%, while we obtained 86%. In most of the crops such as winter triticale, spring oats, potato, green grain spring barley, peas, grass for seed, and permanent grass, their network could not predict these classes (Figure 14).

**Table 3.** Comparison of classification accuracies between our approach and other state-of-art methods. In the table, P and f represent pixel and field-based accuracy, respectively.

| Approach | Sensors (Sentinel) | Number of Classes | Classifier | Classification Accuracy | Our Dataset |
|---|---|---|---|---|---|
| Zhou et al. (2017) [6] P | S1 | 5 | Support vector machine (SVM) | 93% | – |
| Van Tricht et al. (2018) [13] P | S1 and S2 | 12 | Random forest (RF) | 82% | – |
| Mullissa et al. (2018) [22] f | S1 | 13 | FCN-DK3 | 54% | – |
| Ndikumana et al. (2018) [21] P | S1 | 11 | Recurrent neural network (RNN) | 89% (cross-validation five times) | 71% |
| Wei et al. (2019) [23] P | S1 | 4 | U-Net | 85% | – |
| Our approach P | S1 | 15 | FCN-ConvLSTM | 86% | 86% |

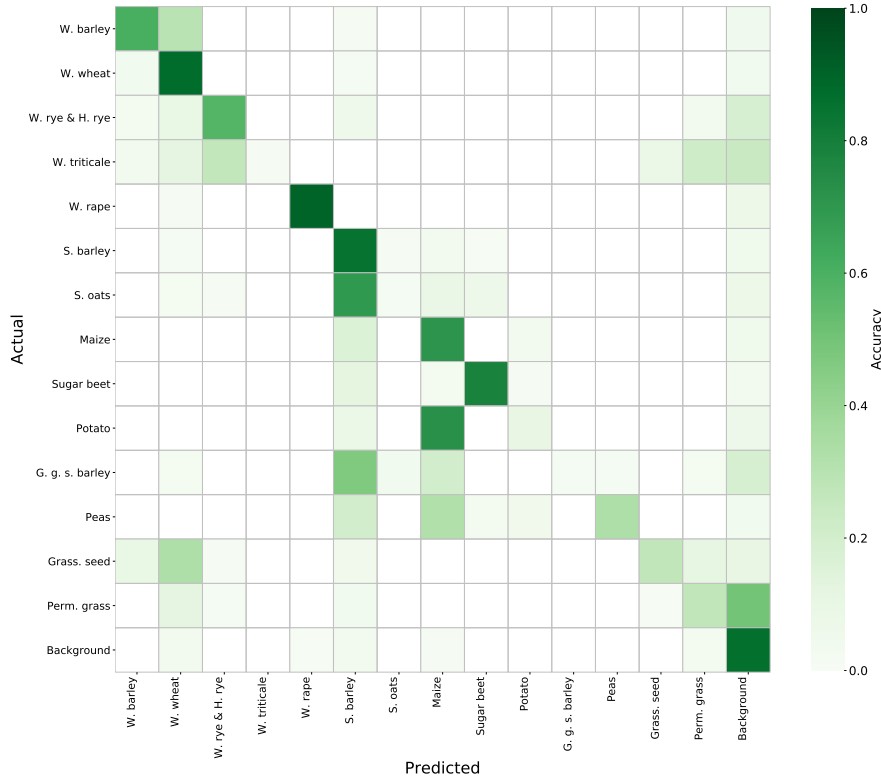

**Figure 14.** Normalized confusion matrix by [21] on our dataset.

## 5. Conclusions

The present research investigated the potential of mapping land cover from Sentinel 1 data by using deep learning technology. A novel network structure was developed for this purpose by combining an FCN and a ConvLSTM network. It must be pointed out that developing a lightweight network, which are only about 64 MB in total, is one of the main advantages of this study. The proposed network can be trained by using C-band radar images and is able to extract spatiotemporal features from them. The average pixel-based accuracy and IoU of the proposed network were 86% and 0.64, respectively. The errors were mostly located at field boundaries. One of the main advantages of the proposed network is the detection of fields that are likely annotated incorrectly in the reference images. One of the important but complex classes in this research is the background, which includes extensive regions including lakes, sea areas, forests, buildings, and roads. The pixel-wise accuracy in this class was 88% despite the complexity of the class. This study shows that the sequential network based on a combination of an FCN and a ConvLSTM network provides an efficient approach for classifying different crop types in time-series radar images.

**Author Contributions:** N.T. was responsible for processing the data. N.T. implemented the deep learning method. R.N.J. supervised the development. All authors took part in writing the paper.

**Funding:** This research is a part of the FutureCropping project funded by Innovation Fund Denmark.

**Acknowledgments:** The work was supported by the FutureCropping project funded by Innovation Fund Denmark.

**Conflicts of Interest:** The authors declare no conflict of interest.

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
