# Peer review of "A Novel Spatio-Temporal FCN-LSTM Network for Recognizing Various Crop Types Using Multi-Temporal Radar Images"

_remotesensing, doi:10.3390/rs11080990_

Round 1

Reviewer 1 Report

Line 16: Correct "C-band"

Line 20: Please mention the full form of IoU.

Line 94-95: On viewing the training areas (red), it is observed that they are spatially restricted to some geographic regions. Please explain the reason appropriately why the specific regions were used to choose red and blue areas as training and validation, respectively. 

Author Response

Dear Reviewer # 1:

Concern # 1: Line 16: Correct "C-band"

Author response: 

Thanks for your suggestion, we have corrected it in the manuscript. Also, we revised the paper once again and corrected some English errors. 

Concern # 2: Line 20: Please mention the full form of IoU.

Author response: 

Thanks for your suggestion. We have replaced it with the full form of IoU in the Abstract of the manuscript 

Concern # 3: Line 94-95: On viewing the training areas (red), it is observed that they are spatially restricted to some geographic regions. Please explain the reason appropriately why the specific regions were used to choose red and blue areas as training and validation, respectively.

Author response: 

Thanks for asking this question. Because of distribution various crops in broad area in Denmark, we required to collect images from sentinel 1 for all classes in the regions which most crop types were growing. Therefore, we needed to collect enough number of fields or samples for all classes for training the network, and we checked the shapefile and then we sent request in the fieldbabel web service to get the images from those interested regions.

Please refer to line 98-100 in the new version of the manuscript, in these lines we have prepared the explanation of why we collected the satellite data from these regions in Denmark.

Reviewer 2 Report

The paper presents a new network approach using Fully Convolutional Network (FCN) and a convolutional LSTM network (ConvLSTM) for image radar classification. This research topic is interesting and fit the scope of this journal. The proposed network is trained using C-band radar images from Sentinel-1, being able to extract spatio-temporal features from various agricultural crops. The advantages of this method seem to be the capability of identifying a high number of classes (which, in my opinion is certainly high: 14 classes) from time-series radar images of large areas.

The introduction section can be improved in some points, but overall, the proposed method is clearly described. The analysis of results is solid and statistically sound.  The manuscript is also written in a clear English with an organized structure. Thus, in my opinion, the paper could be approved for publication in Remote Sensing.

However, I have added some comments that could help the authors to improve the manuscript (presented in no particular order):

-        At some points, the authors group a high number of references to support a particular sentence (e.g. lines 42-44, 54-56). These cases may require a breakdown of the citations. For example, in line 54 what key factors are addressed in each study should be indicated.

-        On the contrary, in other parts of the manuscript further references should be added (e.g. lines 45-53).

-        Lines 70-81 represent the truly literature review which serves as the basis for the study (up to this point the text reads clearly but is merely introductory). Thus, an in-depth literature review concerning “Deep Learning approaches and applications using radar imagery and multi-temporal assessment” should be included to improve this part. I would recommend the authors adding one or two additional paragraphs and further references to expand the information provided in these lines. There are a lot of current references about this topic:

*Niculescu, S. et al. (2018). Application of Deep Learning of Multi-Temporal SENTINEL-1 Images for the Classification of Coastal Vegetation Zone of the Danube Delta

*Mullissa, AG. et al. (2018). Fully Convolutional Networks for Multi-Temporal SAR Image Classification.

*Wei, S. et al. (2019). Multi-Temporal SAR Data Large-Scale Crop Mapping Based on U-Net Model

*La Rosa, et al. (2018). Dense Fully Convolutional Networks for Crop Recognition from Multitemporal SAR Image Sequences.

-        The results should also be discussed based on the new references.

Author Response

Dear Reviewer # 2:

Concern # 1: At some points, the authors group a high number of references to support a particular sentence (e.g. lines 42-44, 54-56). These cases may require a breakdown of the citations. For example, in line 54 what key factors are addressed in each study should be indicated.

Author response: 

Thanks for your suggestion, we have revised introduction part once again. And remove some the references and added some new references based on the reviewer’s comments.

Concern # 2: On the contrary, in other parts of the manuscript further references should be added (e.g. lines 45-53).

Author response: 

Thanks for your suggestion, we added new references in revised version of the manuscript in lines 46 and 52 based on reviewer’s comment.

Concern # 3: Lines 70-81 represent the truly literature review which serves as the basis for the study (up to this point the text reads clearly but is merely introductory). Thus, an in-depth literature review concerning “Deep Learning approaches and applications using radar imagery and multi-temporal assessment” should be included to improve this part. I would recommend the authors adding one or two additional paragraphs and further references to expand the information provided in these lines. There are a lot of current references about this topic:

*Niculescu, S. et al. (2018). Application of Deep Learning of Multi-Temporal SENTINEL-1 Images for the Classification of Coastal Vegetation Zone of the Danube Delta

*Mullissa, AG. et al. (2018). Fully Convolutional Networks for Multi-Temporal SAR Image Classification.

*Wei, S. et al. (2019). Multi-Temporal SAR Data Large-Scale Crop Mapping Based on U-Net Model

*La Rosa, et al. (2018). Dense Fully Convolutional Networks for Crop Recognition from Multitemporal SAR Image Sequences.

Author response: 

Thanks for your suggestion. We looked for this ref. “Deep Learning approaches and applications using radar imagery and multi-temporal assessment” but we could not find any paper with this title. If the reviewers send its link then we will add it as new reference in “Introduction” part. However, we added some the relevant references the reviewers suggested to “Introduction” part. 

Reviewer#2, Concern # 4: The results should also be discussed based on the new references.

Author response: 

Thanks for your suggestion. We have prepared the new version of the manuscript which contains new version of Table 3 line 324 and Figure 14 line 327 and we added new references in the last part of the result and discussion based on reviewer’s comments.
